# Acceptability of PrEP among MSM and transgender communities—Qualitative findings from two metropolitan cities in India

**Satyanarayana Ramanaik**[1]*, **Anju Pradhan Sinha**[2], **Aparna Mukherjee**[1], **Ashwini Pujar**[1], **Kalyani Subramanyam**[3], **Anjali Gopalan**[3], **Reynold Washington**[4,5]

**1** Karnataka Health Promotion Trust, Bengaluru, India, **2** Indian Council of Medical Research, New Delhi, India, **3** The Naz Foundation (India) Trust, New Delhi, India, **4** St. John's Research Institute, Bengaluru, India, **5** Centre for Global Public Health, University of Manitoba, Winnipeg, Canada

\* satya@khpt.org, satya.csp@gmail.com

**Data Availability Statement:** Yes, all data and materials from the this study are available without any restriction. However, to protect respondent

## Abstract

### Background

Global evidence suggests that Pre-Exposure Prophylaxis (PrEP) plays a pivotal role in reducing new HIV-infections among key populations (KP). However, the acceptability of PrEP differs across different geographical and cultural settings and among different KP typologies. Men who have sex with men (MSM) and transgender (TG) communities in India have around 15–17 times higher prevalence of human immunodeficiency virus (HIV) than the general population. The low rates of consistent condom use and poor coverage of HIV testing and treatment among the MSM and transgender communities highlight the need for alternative HIV prevention options.

### Methods

We used data from 20 in-depth interviews and 24 focused group discussions involving 143 MSM and 97 transgender individuals from the two metropolitan cities (Bengaluru and Delhi) in India to qualitatively explore their acceptability of PrEP as a HIV prevention tool. We coded data in NVivo and conducted extensive thematic content analysis.

### Results

Awareness and use of PrEP were minimal among the MSM and transgender communities in both cities. However, on being provided with information on PrEP, both MSM and transgender communities expressed willingness to use PrEP as an additional HIV-prevention tool, to complement inability to consistently use condoms. PrEP was also perceived as a tool that could enhance the uptake of HIV-testing and counseling services. PrEP awareness, availability, accessibility and affordability were identified as determining factors that could influence its acceptability. Challenges such as stigma and discrimination, interrupted supply of drugs and non-community-friendly drug dispensing sites were identified barriers to continuing PrEP.

privacy and other personally identifiable information from the qualitative nature of data, any interested researchers will be able to access the data by writing to corresponding authors Dr. Ramanaik (satya@khpt.org) or Karnataka Health Promotion Trust, Bengaluru, India (khptblr@khpt.org)

**Funding:** This study was funded by the Indian Council of Medical Research (ICMR), New Delhi, India. The funders had no role in study design, data collection and analysis, decision to publish, or preparation of the manuscript."

**Competing interests:** The authors have declared that no competing interests exist.

## Conclusions

Using qualitative data from two Indian settings, this study provides community perspectives and recommendations to stakeholders and policymakers for introduction of PrEP into programs as a prevention tool among MSM and transgender communities in India.

## Introduction

Globally, the number of people newly infected or dying due to HIV and AIDS have reduced in recent years. Yet, with a global HIV prevalence of 0.8%, an estimated 37.9 million people were living with HIV in 2018 [1, 2]. Low and middle-income countries bear most of this burden. India has the third-largest HIV epidemic in the world, with 2.3 million people estimated to be living with HIV [3]. An estimated 63000 new HIV infections occurred in 2021, a 37% decrease since 2010. AIDS-related mortality is estimated at 4.43 per 100,000 population in 2019, a decline from its peak of 25 per 100,000 during 2004/05 [4]. Although India's HIV prevention, testing and treatment efforts have resulted in substantial declines in HIV prevalence, the coverage of key populations with critical interventions remains sub-optimal [5]. Key populations (KPs) including female sex workers (FSW), men who have sex with men (MSM), transgender communities (TG), and persons who inject drugs (PWID) remain vulnerable to HIV infection. Interventions with KPs require a more nuanced understanding of their health needs and their socio-environmental contexts.

In India, MSM and the transgender community are at greater risk of HIV infection compared to the general population [6]. HIV prevalence among MSM (3.3%) and transgender community (3.8%) is about 15 to 17 fold higher than the general population (0.22%) [3, 7]. Despite national efforts to ensure their health rights and wellbeing, they continue to face substantial structural and interpersonal barriers while accessing HIV prevention and care services [8].

MSM and transgender communities are biologically, socially, and structurally more vulnerable to HIV. The taboo against their gender identity, the social stigma and discrimination that these populations face and their discrete behaviors within society make it challenging to design and implement HIV prevention interventions that enable access to essential HIV prevention services [9, 10]. Moreover, studies show that inconsistent condom use, frequent alcohol use, experiences of sexual violence and engagement with multiple concurrent partners among both the MSM and transgender communities raise their risk for HIV [11, 12]. Experiences of sexual violence among MSM and TG communities puts them at a much higher risk of HIV [13, 14]. It is therefore essential to provide other forms of prevention that can address these key populations' risk context in a manner that is culturally appropriate and customized to their concerns. One such option is the use of Pre-Exposure Prophylaxis (PrEP) as a HIV prevention method. PrEP is a valuable opportunity to implement effective HIV prevention interventions among MSM and transgender communities [1]. Recent trials and feasibility studies on PrEP around the world have shown promising results among the transgender communities, MSM, and Female Sex Workers (FSW) [15–17] with evidence pointing out that consistent and correct use of PrEP can significantly reduce the chances of HIV infection [18].

It is therefore important to explore whether KPs are willing and able to adopt and adhere to prescribed regimens of PrEP. Literature on the acceptability of PrEP among MSM and transgender communities in India is limited. It was in this context that the Indian Council for Medical Research (ICMR) task force on HIV Prevention commissioned a study, that was implemented by St. John's Research Institute in partnership with Karnataka Health Promotion Trust (KHPT), Bengaluru and the Naz Foundation, Delhi. The study aimed to understand the

barriers and enablers to acceptance of PrEP as a prevention tool among the MSM and transgender community. This study also explored community perceived solutions and conditions in two culturally different contexts that would be required for rolling out PrEP programs in India.

## Methodology

### Study design and sample

This qualitative study explored the perspectives of MSM and transgender communities to their acceptability of PrEP. Focus Group Discussions (FGDs) were used to explore the participants knowledge and perspectives on PrEP, feasibility for regular use, potential challenges with regular use, and processes by which programs could popularize or roll out PrEP in order to enhance acceptance by the community. In-depth interviews (IDI) were also conducted to gain a deeper understanding of more specific and personal issues such as condom use, HIV testing, and interpersonal dynamics within the MSM and transgender communities in the two study sites in India. The study was conducted in two major metropolitan cities, New Delhi and Bengaluru, between July 2017 and June 2018. The Naz Foundation, New Delhi and the Karnataka Health Promotion Trust (KHPT), Bengaluru, were involved in study implementation.

Purposive sampling was used to recruit eligible participants. Eligible participants included distinct types of MSM and transgender communities. MSM refers to self-identified gay men (western acculturated), *kothis* (men who tend to be the receptive male partner in anal and oral sex and typically have more feminine mannerisms), *panthis* (men who tend to be the insertive male partner in anal and oral sex), and *double-deckers* (men who are both receptive and insertive partners) [19]. Transgender women were those who reported male at birth but self-identify as women or female. Transgender community are broadly categorized as transgender Hijras and transgender non-Hijras [20, 21]. All participants were in the age group of 18–45 years and provided written consent to participate in the study. In total, 20 IDIs (Bengaluru 11 and Delhi 9) and 24 FGDs (Bengaluru 12 and Delhi 12) with 220 total participants were completed in about six months, starting from September 2017 to February 2018. Local community-based organizations who were implementing targeted intervention programs under the National AIDS Control Program, were involved in mobilizing participants. Both FGDs and IDIs were conducted in places where the participants felt safe and comfortable and included Community Based Organization/Non-Government Organization (CBO/NGO) offices and field-level clinics.

Subsequent to posing the initial questions on PrEP knowledge, interviewers provided basic PrEP information to all participants, including the composition of PrEP, the mechanism by which PrEP works for HIV prevention, the dosage required, its availability and cost in India, in simple non-medical language. The information provided participants with knowledge to respond to questions on the usefulness and acceptability of PrEP by the community.

### Study tools and data analysis

We used independently developed and pre-tested semi-structured interview guides for both IDI and FGDs. The tools contained questions to assess the participants' knowledge and risk perception of HIV/AIDS, views and perception on PrEP, and knowledge and utilization of preventive interventions that were currently offered within the targeted intervention programs. Extensive probes were used to elicit additional information on each theme. In addition to participating in the interviews, we asked participants to complete HIV service mapping activities. These intervention maps provided us with a wealth of information about the population's knowledge of services that were accessible to them and the geographical or social

implications of current service locations that could potentially offer PrEP services. We trained interviewers from both the sites over two days. Most interviewers had previous experience of conducting qualitative studies. The training included research ethics, study objectives, qualitative research methods, and aspects on PrEP.

The field teams transcribed all 24 FGDs and 20 IDI from Bengaluru and Delhi sites. Field supervisors verified the verbatim (audio to regional languages) transcription documents for accuracy and completeness. Subsequently, the team translated all transcripts into English. We used NVivo-11 to assist qualitative data management and analysis [22]. We held a consultation workshop with field investigators from both sites in order to develop the analytical framework. The thematic content analysis was supplemented by constant comparison and deviant case analysis techniques.

### Ethical considerations

We obtained written informed consent from all study participants for their voluntary participation and audio recording of the interviews. We maintained anonymity by using a unique study ID to distinguish individual participants in the study. While presenting our findings in this paper, we use pseudonyms. The Institutional Ethics Committee of St. John's Medical College and Hospital, Bengaluru (Ref # 92/2014) approved this study. In consideration of the potential risks of disclosure of the study subject's identity, we took extra measures to secure and de-identify the documents. During the training, we briefed the data collectors about the sensitivity of the subject and the necessity for discretion and privacy regarding personal identifying information.

## Results

We present awareness, acceptability, concerns around use and community's suggestions on the most appropriate methods to roll out PrEP programs.

### Profile of the participants

A total of 143 MSM and 97 transgender individuals were involved across both sites. The socio-demographic characteristics of the participants are listed in Table 1.

### Awareness about PrEP

For a majority of the participants, the first introduction to PrEP was during this study.

*Is PrEP useful for HIV positive people or only HIV negative people? How soon will PrEP provide protection? Can we take the PrEP medicines in conjunction with other medications? (Sandesh, MSM, 29 years, Bengaluru).*

Their limited information had been previously gathered during informal interactions, rather than formal PrEP educational sessions.

*I have only heard about the tablet's name. Nobody in the community has explained or told me about this (Anand, MSM, 30 years, Bengaluru).*

In both Delhi and Bengaluru sites, transgender communities displayed a better understanding of PrEP as compared to the MSM group and sources of information included peer networks, internet and social media, or non-government organizations. In Bengaluru, transgender individuals who had heard of PrEP were not sure about its availability in India, its

**Table 1. Sociodemographic characteristics of MSM and transgender communities in IDI and FGD (n = 240).**

| Characteristics | MSM | transgender community | Total n = 240 (%) |
|---|---|---|---|
| | n = 143 (%) | n = 97 (%) | |
| **Age** | | | |
| >25 | 44 (30.76) | 43 (44.32) | 87 (36.25) |
| 26–40 | 75 (52.44) | 49 (50.51) | 124 (51.66) |
| 40+ | 24 (16.78) | 5 (5.15) | 29 (12.08) |
| **Highest level of education completed** | | | |
| No formal education | 11 (7.69) | 7 (7.21) | 18 (7.05) |
| Primary | 24 (16.78) | 21 (21.64) | 45 (18.75) |
| High school | 56 (39.16) | 39 (40.20) | 95 (39.58) |
| College | 52 (36.36) | 30 (30.92) | 82 (34.16) |
| **Primary Identity** | | | |
| Kothi | 68 (47.55) | 0 | 68 (28.33) |
| DD | 58 (40.55) | 0 | 58 (24.16) |
| Bi-sexual | 7 (4.89) | 0 | 7 (2.91) |
| Panthi | 6 (4.19) | 0 | 6 (2.05) |
| Gay | 4 (2.79) | 0 | 4 (1.66) |
| Hijra Identified | 0 | 94 (96.90) | 94 (39.16) |
| Trans Identified | 0 | 3 (3.09) | 3 (1.25) |
| **Marital Status** | | | |
| Married | 53 (37.06) | 10 (10.30) | 63 (26.25) |
| Unmarried | 86 (60.13) | 84 (86.59) | 170 (70.83) |
| Divorced/separated | 4 (2.79) | 3 (3.09) | 7 (2.91) |
| **Main Occupation** | | | |
| Sex work | 20 (13.98) | 25 (25.77) | 45 (18.75) |
| Daily wage labour | 45 (31.46) | 31 (31.95) | 76 (31.66) |
| Works at CBO | 14 (9.79) | 5 (5.15) | 19 (7.91) |
| Private | 20 (13.98) | 9 (9.27) | 29 (12.08) |
| Business | 6 (4.19) | 0 | 6 (2.05) |
| Unemployed | 5 (3.49) | 0 | 5 (2.08) |
| Toli Badhai/begging | 0 | 15 (15.46) | 15 (6.25) |
| Not disclosed | 33 (23.07) | 12 (12.37) | 45 (18.75) |

cost, how it was to be consumed or its effect on HIV prevention. Few participants from both the communities misunderstood PrEP to be PEP (post-exposure prophylaxis).

The researchers responded to questions before further probing on perspectives towards PrEP acceptability.

## How can PrEP benefit your community?

In both Delhi and Bengaluru, the transgender and MSM participants felt that PrEP could be very beneficial as it could provide "double protection" to them. Both groups also found it useful in instances when the condom slips off or tears.

*We can control HIV/AIDS by consuming the PrEP tablets. Similarly, we could control STI's by using condoms externally. Both are good to safeguard our health. Sometimes, if we are going ahead in our sexual interaction even without condoms also, we would feel safe and confident. . . we will not be infected with HIV if the condom breaks/pops/slips off. . . (Rupa, transgender individual, 29 years, Delhi).*

The MSM population felt that PrEP had an additional benefit to them as medicines could be consumed without fear of being judged by others, whereas carrying condoms to the home could raise suspicions from family members.

*Female or male, they can't carry the condoms in their pocket or bags. If someone in our household came to know that we have condoms, then there will be fights in the homes. If there are tablets, then people won't suspect much. . .. If such kind of a tablet is introduced, then it will be good. (Sourav, MSM, 34 years, Delhi).*

*If we start consuming pills on a daily basis, the family members could inquire about the reason for consuming those tablets. Mothers would be always the first person to inquire about them. Whether for fever or any other health ailments. We might be able to keep it confidential from anybody but to keep it so private from mother or others would be very difficult. We might still go ahead in saying something like vitamin tablets, etc., but we can't do so for a longer period. (Janardhan, Bisexual, 48 years, Bengaluru).*

Few participants expressed a view that PrEP could reduce the frequency for HIV testing. The transgender communities reported that PrEP could be particularly relevant when they desired more intimate relationships with their partners or when their clients did not agree to use condoms. During forced sex or coercion, PrEP could offer them protection from HIV infection.

*Sometimes telling them (clients) to use condoms is difficult and in such situations, taking this pill gives more freedom from tension. (Preeti, transgender individual, 26 years, Bengaluru).*

Non-panthi MSM and transgender individuals perceived that Panthis were largely responsible for HIV transmission. They would often coerce or trick partners into abandoning condoms. Panthis have been known to take condoms off before or during intercourse, without their partner's knowledge. They threaten partners who insist on condom use, or intentionally damage condoms such that they tear during intercourse. Their stories suggest that condoms are not an effective HIV prevention option among non-panthi MSM and transgender individuals. Male Sex Workers (MSW) in the group are often offered more money to forgo condoms. Poverty and deprivation often force them to concede to their customer's demands. Clients were known to become violent if MSW refuse sex without a condom and the interactions usually end up with violence, including rape. These power dynamics within the community adversely influence the effectiveness of condoms for HIV prevention. Participants expressed that they would be at lower risk of HIV infection with PrEP.

*I know that our community participates in sex work and carries condoms. Sometimes in hamams they have more than 20–30 clients a day. It is exhausting and they may not have much patience to wear the condom properly, to check whether there is air inside or not and she may be not much cautious about the breakage and all so, therefore, the tablet will be helpful. (Preeti, transgender individual, 26 years, Bengaluru).*

One more major identified barrier to condom use was alcohol and substance abuse. Many participants stated that when they were under alcohol influence, it became difficult to use a condom. MSW shared that clients who drank or took drugs were often vehemently opposed to using condoms.

*Those customers/clients who do approach us under the influence of alcohol/drinks, we find very difficult. We find it very difficult for those who are not in their true senses because of their addiction of drugs/narcotics. Whatever ways we try to convince them, it would be futile and they don't listen to our request. (Keerthi, transgender individual, 36 years, Bengaluru).*

Also, the issue of using inappropriate lubricants was reflected in both MSM and transgender communities' interviews. Many participants reported difficulty in finding gels or other approved sexual lubricants. In the absence of gel, participants reported that community members use substitutes such as cooking oil, lotion, or coconut oil. These substitutes are preferred because they are easily available, affordable, and do not suggest 'socially tabooed' behavior. However, oil-based solutions are not appropriate for use as a sexual lubricant. Many of the participants knew that oil-based lubricants compromise the integrity of condoms, leading to condom breakage or tears during intercourse, but yet admitted that these substitutes are sometimes used even without a condom to reduce friction and pain and to increase pleasure during sexual intercourse.

Most of the MSM and transgender communities were keen that PrEP messaging should be communicated to other high-risk communities including FSWs and PWID. The network between high-risk communities is complex. Therefore, providing information and education on PrEP to MSM and transgender communities alone may not be sufficient. It is important to consider all high-risk communities while designing PrEP rollout programs in the country. Educating the general community, especially the youths, will not just help to protect them from HIV, but will also reduce stigma and discrimination attached to the key populations in India.

*I think all communities will benefit. Everyone wants to be safe. I understand why the focus is on my community, my community is at high risk with multiple partners, into alcohol and they maintain secrecy. This way they are more at risk of getting infected by HIV (Kajal, transgender individual, 34 years, Delhi).*

Our results indicate that both in Delhi and Bengaluru and among both MSM and transgender communities, a need to have alternative HIV protection methods such as PrEP was useful. However, mild differences existed in the perceived reasons for PrEP use. The transgender communities largely perceived that PrEP would offer them HIV protection despite their compromised decision-making and frequent encounter of violence during sexual interactions. However, among the MSM, their major concern was secrecy from the family and society about their sexual identity, with PrEP offering an alternative to the challenges around condom-use within their homes.

## Concerns over the use of PrEP

Results from both sites, and across typologies, elicited different factors that could impact the acceptability of PrEP. All participants expressed their concerns about making PrEP easily and consistently available in India. They pointed out that there were major interruptions with regular supply of condoms and antiretroviral treatment (ART) that occurred with changing program leadership and supply chain issues. Even large hospitals often faced shortage of HIV prevention and treatment materials and supplies. Often in the past, ART drugs were withdrawn from their communities, when different political parties came into power. They were concerned that similar lack of consistent availability could arise with PrEP and that this could lead to drug-resistant strains developing within the community.

*Sometimes the present government may start providing the tablets. Then if we have the next government they may stop or not give it on time. (Rahim, Kothi, 27 years, Bengaluru).*

The high costs of PrEP was also expressed as an important concern by both the MSM and transgender communities in both the study sites. Many participants said, a large number of MSM and transgender communities engage in sex work to fulfill their financial needs and though they understand they are at increased risk to contract HIV, the potential cost associated with using PrEP was a deterrent to its use. Participants were also concerned that the additional lab tests prior to and during PrEP would be exorbitant in cost and beyond their means.

*Earning money is itself very tough . . . therefore if they are being distributed free of cost it would be preferable. (Ranga, Kothi, 32 years, Bengaluru).*

Many members of the MSM and transgender communities spoke about facing stigma and discrimination when they venture into public spaces and in health care settings. Many participants reported being treated disrespectfully within government hospitals.

*I was facing a lot of problems when I go to health care facilities. Lots of discrimination, verbal discrimination, and nonverbal discrimination. I become very emotional and I suffered from depression because of it. I am afraid when I go to public places. (Preeti, transgender individual, 26 years, Bengaluru).*

MSM, who were not outwardly identifiable, stated that they often felt judged when they divulged that they engaged in risky behavior with other men. Transgender communities, who are unable to hide their diversity from what is considered sociably acceptable, stated that they face discrimination as soon as they enter facilities. When they are assigned to a physician or a nurse, the provider often refuses to touch them or to do a thorough or complete physical or clinical examination. Some transgender communities shared that they often have longer waiting times than other patients and are often sent away from facilities without receiving services, or given rushed and incomplete services in an effort to dismiss them quickly. They also reported instances of being sent to various other facilities for simple or routine tests. Participants were concerned about how these issue would be resolved when PrEP is made available. For these and other reasons, participants prefer to go to the private hospitals, which are nearer to their residence and open for longer hours. They have built a better rapport with private hospital doctors, which also have functional equipment and regular supplies to perform all tests in one place. Another concern raised was the side effect of the medicines and the risk associated with prolonged use.

*When you told me this tablet has to be consumed every-day, I was only thinking of side effects and drug resistance. (Pradeep, MSM, 30 years, Bengaluru).*

MSM expressed concerns over additional side effects of PrEP when taken along with alcohol. Transgender communities were concerned that PrEP could negatively affect their bone density and create bone-related problems as they age, as many of them are on Hormone Replacement Therapy, which has a similar effect on bone density. Transgender communities in Bengaluru, also shared their concerns about side effects of PrEP, as many of them were already taking medicines for diabetes and blood pressure. They insisted that PrEP should be customized such that it is indicated only whenever they need it, that is before the sexual encounter.

All participants stated that many people may confuse PrEP with ART because both are daily regimens. This confusion can cause PrEP users to experience discrimination and community isolation, similar to the experience of persons living with HIV (PLHIV) who are on ART. This issue prompted participants to ask about possibilities of intermittent PrEP, or formulations of PrEP in the guise of a general health tonic, or for other PrEP formulations that would not necessitate daily consumption.

*When we take them every day, they [family member] will relate it to ART and suspect that we are having HIV. . . so our people start looking at us differently. (Preeti, transgender individual, 26 years, Bengaluru).*

Hijra transgender communities were concerned that their frequent travel could pose barriers to PrEP adherence. Hijras travel a lot to perform religious ceremonies and sometimes visit their rural homes and feared that they could run out of PrEP supplies and would be unable to get a timely refill in their villages. They expressed apprehensions that missing PrEP doses would lead to drug-resistant strains. They also shared that Hijras and Male Sex Workers' late working hours and alcohol consumption could pose barriers to adherence.

*Some of our people can't consume it daily because they drink alcohol. . ..Our community thinks that those who are taking the tablet daily, they suffer AIDS or TB, or any other serious infection. (Rehana, transgender individual, 30 years, Bengaluru).*

Many MSM and transgender individuals participate in sex work intermittently. They said that it would be difficult for them to plan ahead and start the medicine 6 days prior to initiation of sexual activity and therefor preferred condoms as they were readily available and familiar to use.

## How can PrEP be made available to MSM and transgender communities in India?

We enquired what community-friendly measures could be used to popularize and roll out PrEP in India. The first and foremost priority mentioned was to educate MSM and transgender communities on PrEP. All participants expressed interest to learn more about PrEP and were willing to share their knowledge with their peers. Discrete education methods using social media platforms like SMS, WhatsApp or Facebook were preferred options for the privacy and safe space that they offered.

*We should advertise, through public media and social media like Facebook and WhatsApp, everywhere we should spread the messages (Uma, Kothi, 22 years, Bengaluru).*

Transgender communities in Delhi also suggested that information could be provided at Hamams (their workplace). Participants who were in favor of learning in a safe space also suggested that NGOs and CBOs who work with these communities could begin PrEP education campaigns. However, married and bisexual participants pointed out that they rarely visit the CBO or NGO offices.

*If the initiations are taken through our CBO, then the majority of our community people would be getting the information by verbal communication in more effective ways. There won't be any problem in passing on the information with enthusiasm and happiness. (Girish, Panthi, 40 years, Bengaluru).*

*Similar to other ways of sharing information through radio, television, from friends working in NGOs, etc. (Manish, Panthi, 30 years, Delhi).*

Some participants were in favor of widespread PrEP education campaigns that included general population and youth. Information, education and communication (IEC) materials such as flipcharts, brochures, pamphlets may be very useful in schools and colleges, government hospitals (only Kothis mentioned this), mobile platforms, and social media. Street-plays, drama and education campaigns could also reflect on PrEP.

With regards to affordability, most of our participants suggested that PrEP should be offered free of cost, just like free condoms that the government offers. This could be the government incentive to encourage PrEP use by the MSM/ transgender community. However, few opined that a nominal price could be charged to discourage its misuse, as free supply diminishes the perceived value of medicines and are liable to be discarded. Noteworthy is the fact that many of the participants in this study recruited were from a lower socioeconomic status. Most participants suggested that if there is a cost, it should be low enough to be affordable, at about Rs 5-10/- per month. Those from upper economic strata were willing to pay up to Rs 500/- per month.

*We earn by begging; we expect it to be in the range of Rs 5/- to Rs 10/- for a month. In that case, only, our community could afford the price. Or else we will end up discontinuing the medicine. (Reshma, transgender individual, 29 years, Bengaluru).*

*If you distribute free of cost, people could lose importance and someone could misuse or waste them also. . . If they pay, they would show some concern about them. (Kiran, Bisexual, 30 years, Bengaluru).*

The third issue was accessibility. All participants agreed that PrEP should be made available in facilities that MSM and transgender communities can comfortably access. Suggested locations included NGO and CBO offices, urban health clinics (Mohalla clinics), private hospitals that serve and empathize with MSM and transgender communities, and other self-service condom distribution points that are currently accessible to this vulnerable population. Few transgender individuals in Delhi reported that they also prefer to receive it from a petty shops like paan shops, where they could simply pick it up in the disguise of purchasing some other item.

*This tablet should be made available everywhere in the NGOs and clinics like mohalla because people avoid going to big hospitals as they think where will they wander around in those hospitals asking for tablets. So, if it's available at Mohalla clinic it will be easy for us to take them. (Rohit, Gay, 32 years, Delhi).*

Participants also reiterated that importance of sensitizing healthcare professionals and the general community to address the stigma and discrimination attached to the key populations in India.

*When I go to the government hospital they ask us, "being a man you have sex with a man, are you not ashamed of it?" They ask like this. After seeing all these, I don't like to go there. (Hema, transgender individual, 37 years, Bengaluru).*

Finally, the study subjects was asked about their perceptions on PrEP as compared to condoms. Both positive and negative responses were received. A majority of participants expressed a need for both condoms and PrEP. Both MSM and transgender communities

**Table 2. Comparison between perspectives of MSM and transgender communities on PrEP acceptability.**

| | MSM | TRANSGENDER COMMUNITIES |
|---|---|---|
| **Need (for alternative HIV preventive-PrEP)** | Difficult to carry condoms home (*fear of suspicion among the family members, esp. in younger MSM or unmarried MSMs*) | Difficult to negotiate with clients/partners for condom use (*Dominating partner, excessive alcohol use, the demand of intimacy*) |
| | Can't buy lubricants | Use of inappropriate lubricants |
| | Preference for intimacy | Preference for intimacy |
| **PrEP knowledge** | A large number of MSM have poor knowledge of PrEP | Few transgender individuals were better informed about PrEP |
| **Storage** | Can't store medicines at home | May forget to carry while on travel |
| | Taking medicines regularly may create suspicion among the family members | Cannot always plan their sexual encounters beforehand due to the nature of the job |
| **Side effects** | General concerns about the side effects of the PrEP drug | Few were concerned about the side effects of the PrEP medicine on bone density, as they undergo hormone replacement therapy which already poses a risk for decreased bone density. |
| | | Continuous use may lead to drug resistance |
| | | Side effects of the medicine when taken with medicines for multiple comorbidities like diabetes and BP |
| **Testing** | Discomfort with HIV testing as pre-requisite to initiate PrEP | Discomfort due to their experiences of stigma and discrimination, as a result of which they do not access regular HIV testing |
| **Cost** | Preferred a low cost ranging from Rs 30-500/- per month | Preferred it for free, or a nominal price. Rs 5-10/- per month |
| **Availability** | Preferred through CBOs or NGOs. Few also suggested it should be available at the pharmacist or 'paan' shops or on internet e-portals | Preferred only through CBOs |
| **Usage** | Preferred to combine both condoms and PrEP | Preferred to combine both condom and PrEP |
| | Preferred a syrup or tonic or one tablet taken on 'event' basis | Preferred one tablet taken as an emergency pill |

admitted that their lifestyle and occupation puts them at increased risk for contracting Sexually Transmitted Infections (STIs) and HIV. PrEP can offer protection against HIV when condoms cannot be used. In addition, condoms protect against many STI. Few participants preferred condoms to PrEP stating that they were already familiar with condom use and knew where to access them.

In Table 2 we summarize the similarities and differences in perspectives of MSM and transgender communities regarding their acceptability towards PrEP.

## Discussion

Our study explores the perceptions on the use of PrEP as a HIV prevention method among MSM and transgender community in two cities, one each from north and south India. The study also explored opinions for the configuration of a potential program roll-out of PrEP that would enhance the acceptability to these communities. In recent years, there has been an increase in the number of studies on PrEP acceptability among transgender women and MSM in India [11, 12]. However, not all studies on PrEP acceptability have described and compared the differences in perceptions or reasons for these perceptions between MSM and transgender communities.

The study is unique as it highlights the barriers to access for HIV prevention services that transgender communities and MSM face. The study also provides in-depth insights into targeted strategies that can be adopted to ensure that PrEP acceptability and use is optimized, when a PrEP program is rolled out.

The generic version of PrEP was approved for use in India in 2016. Yet, it is not available through the public healthcare system or at a subsidized cost in the private sector. Presently, few private practitioners prescribe PrEP and this is available with physical and online pharmacies. The estimated number of PrEP users in India ranges between 1500–2000 [23]. Anecdotal

evidence suggests that most current users of PrEP are educated MSMs, belonging to the higher socioeconomic groups [11].

Currently, HIV targeted interventions focus on condoms as the sole method for prevention of HIV among MSM and transgender communities in India. The test and treat option is implemented and treatment can reduce the risk of HIV transmission, but substantial proportions of MSM and transgender communities do not regularly access HIV testing. This study also found that a large number of participants were not able to use condoms during sexual interactions, similar to other studies [24]. Condom use is often compromised because of inability to negotiate its use, or the need for intimacy with their partner [12, 19]. The feminine identity, poverty and dependence on sex work as a major source of livelihood among the transgender communities and MSM, create a hierarchy in power dynamics with their clients, who are often masculine with greater power in decisions pertaining to condom use. Moreover, misconceptions like double bagging and the use of inappropriate lubricants raise other challenges with condom use [14, 25]. Under such circumstances, most participants expressed that PrEP could be a useful complementary method of HIV prevention. However, very few participants were aware about PrEP and its use, with awareness levels slightly higher among the transgender communities. For many of the participants, the first introduction to PrEP was through this study. Most of the participants were excited by the possibilities that PrEP could offer and expressed surprise that PrEP information was not already circulating within the community. A conclusion reached was that PrEP awareness campaigns are crucial to its acceptability. Other studies have also found large proportions (93 percent) of participants were not aware of and had never heard about PrEP [26]. However, once informed about PrEP, their willingness to use PrEP was extremely high (99%) [26]. Both MSM and transgender community asserted that they need complete information regarding side effects, long-term health complications, interactions with other medications, drug resistance, costs, and pre-requisite tests. Potential side effects of PrEP remain a major concern among the MSM and transgender communities, similar to other studies [27]. Other studies have also pointed out that more trials on intermittent and long-term PrEP are necessary to rule out the possible side effects [28, 29].

Similar to the findings of Chakrapani et al, MSMs in our study also raised concerns about storage of PrEP drugs at home [11]. Transgender communities, on the other hand, expressed apprehension because of their challenges in access when they regularly travel to other places for work. Other studies too have shown that PrEP acceptability depends on 'event-driven' needs and is preferred where one of the partners is HIV infected and daily intake was one of the biggest obstacles to PrEP use [16, 17].

Concerns around the cost of PrEP are real not only in our study but also in middle and high-income countries [30, 31]. Chandhiok et al noted that cost was a major barrier to initial commitment, while an increased willingness to pay for PrEP was seen among younger age groups [32]. Our study also highlighted the need for a steady supply of the PrEP drug similar to a study by Gomez and Harris [33].

A pre-requisite of initiating PrEP is to undergo regular HIV testing, to confirm a negative status. This implies that HIV testing services, PrEP and treatment should be supportive towards key populations. Similar to other studies, our study too points out that stigma and discrimination remains a significant barrier preventing sexual minorities, especially the transgender communities from accessing healthcare services. Eliminating discrimination in healthcare services is one of the priorities towards ensuring their rights [34, 35]. A better understanding of the various factors that influence sexual risk behaviors and disclosure practices among HIV-positive MSM and 'hijras' can help counselors, physicians, and peer educators in providing appropriate information and nonjudgmental counseling and clinical services. This would also

help the policymakers to design specifically tailored approaches within HIV prevention programs, as recommended by others [11].

Involving CBOs in the distribution of PrEP is an effective way to increase acceptance [21]. Similarly, programs designed to reduce HIV incidence among the key population should be centered around community empowerment and basic human rights [36–38]. The 'PrEP continuum of care' theory by Kelley et al., 2015 emphasizes awareness and subsequent willingness to initiate PrEP among those at highest risk for HIV infection as a necessary first step [31]. Various theories of acceptability, such as the 'Health belief model' that considers the perceived benefits and the perceived barriers, and the 'diffusion of innovation theory' that suggests use by a small proportion will enhance use by larger numbers of peers, reinforce this option of involving CBOs for PrEP programs. CBOs are already actively involved in addressing basic needs of transgender communities. Public health professionals and healthcare providers often expect transgender communities to account for their sexual and reproductive health when they are denied basic human rights including shelter, secure jobs and the ability to provide food. One cannot expect that they would prioritize purchasing PrEP over meeting these primary needs.

This study has some limitations. Firstly, the participants were asked to share their opinion on PrEP without giving them the product. None of the participants had ever used PrEP and their responses were based upon information provided by the research team. It is possible that information provided by research team members differed based on the setting and duration of interactions. Secondly, as a qualitative study with a relatively small sample size, generalizations certainly cannot be made beyond the participant groups.

## Conclusion

This study demonstrates that PrEP is acceptable among MSM and transgender communities in India. However, conditions that ensure its regular availability and accessibility should be in place. We did not find much difference in acceptability between the two sites, but the barriers and challenges are different for transgender communities and MSMs. The strategies to advocate PrEP use should thus be contextualized for each community. PrEP education campaigns are necessary not only among MSM and transgender communities, but also for health care providers and the general population. PrEP accessibility increases when offered in places where MSM and transgender communities can access services without prejudice. Cost is a significant factor determining the acceptability of PrEP among MSM and transgender communities. PrEP needs to be 'de-medicalized' and offered in non-clinical settings, much like condoms. PrEP may be offered within a 'package of services', rather than as a stand-alone intervention, as it is complementary to existing prevention initiatives and is especially useful among those with high-risk behavior who have challenges to consistently use condoms. However, when the concerns and benefits of PrEP are evaluated from the perspectives of the transgender communities and MSMs, it is clear that unless all concerns related to uninterrupted availability in a stigma free environment and at an affordable cost are addressed, the likelihood of its acceptance remain slim.

## Supporting information

**S1 File.**
(DOCX)

## Acknowledgments

We thank the member of MSM/transgender communities from both Bengaluru and Delhi for participating in the study and providing valuable information. We also acknowledge the

support received from the Karnataka State AIDS Prevention Society and Community Based Organizations such as Samara, Samara society and Payana, Bengaluru and Community Empowerment Trust, Basera Samajik Sansthan, New Delhi. We acknowledge the support by Dr. Swaroop N, Ms. Sunitha BJ, Dr. Michael Baburaj, Mr. Sijil Joseph, Ms. Piyali Acharya, and Mr. Sarabjeet Singh in data collection and translation. Also, thanks to Ms. Suchandrima Chakraborty, Ms. Frost SD, Ms. Kavitha DL, and Dr. Ravi Prakash for their contribution to data analysis and initial draft report. We also extend our utmost gratitude to the team at St. John's Research Institute, Bengaluru. Finally, we thank the field staff and the administration/ finance teams of Karnataka Health Promotion Trust, Bengaluru, and The Naz Foundation (India) Trust, New Delhi for their hard work and timely support for this study.

## Author Contributions

**Conceptualization:** Satyanarayana Ramanaik, Anju Pradhan Sinha, Kalyani Subramanyam, Reynold Washington.

**Data curation:** Satyanarayana Ramanaik, Aparna Mukherjee, Ashwini Pujar.

**Formal analysis:** Satyanarayana Ramanaik, Aparna Mukherjee, Ashwini Pujar.

**Funding acquisition:** Anju Pradhan Sinha, Reynold Washington.

**Investigation:** Satyanarayana Ramanaik.

**Methodology:** Satyanarayana Ramanaik.

**Project administration:** Ashwini Pujar.

**Resources:** Reynold Washington.

**Software:** Satyanarayana Ramanaik, Aparna Mukherjee.

**Supervision:** Satyanarayana Ramanaik, Kalyani Subramanyam.

**Validation:** Satyanarayana Ramanaik, Kalyani Subramanyam.

**Visualization:** Satyanarayana Ramanaik.

**Writing – original draft:** Satyanarayana Ramanaik, Aparna Mukherjee.

**Writing – review & editing:** Ashwini Pujar, Kalyani Subramanyam, Anjali Gopalan, Reynold Washington.

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
