## [Decision Letter · Decision Letter 0]

18 Aug 2021

PONE-D-21-09058

Acceptability of PrEP among MSM and Transgender - qualitative findings from two metropolitan cities in India

PLOS ONE

Dear Dr. Ramanaik,

Thank you for submitting your manuscript to PLOS ONE. After careful consideration, we feel that it has merit but does not fully meet PLOS ONE’s publication criteria as it currently stands. Therefore, we invite you to submit a revised version of the manuscript that addresses the points raised during the review process.

Please address the comments and suggestions of the reviewers and submit a revision. Improved clarity of sampling and analysis methods are especially needed.

We look forward to receiving your revised manuscript.

Kind regards,

Dawn K. Smith

Academic Editor

PLOS ONE

Journal Requirements:

2. Please change potentially stigmatizing or outmoded terms such as "the transgenders" to or "transgenders" to currently accepted terminology such as "transgender people" or "transgender communities.

Reviewers' comments:

Reviewer's Responses to Questions

**Comments to the Author**

1. Is the manuscript technically sound, and do the data support the conclusions?

Reviewer #1: Partly

Reviewer #2: Yes

2. Has the statistical analysis been performed appropriately and rigorously? 

Reviewer #1: N/A

Reviewer #2: No

3. Have the authors made all data underlying the findings in their manuscript fully available?

Reviewer #1: Yes

Reviewer #2: No

4. Is the manuscript presented in an intelligible fashion and written in standard English?

Reviewer #1: No

Reviewer #2: Yes

5. Review Comments to the Author

Reviewer #1: The study in the manuscript is a very relevant topic given the concentrated HIV/AIDS epidemic in India. It has the potential to inform the policies towards the framing of PrEP interventions. However, the authors may like to review the following to enhance its readability and rigors.

1. Please use the data on the latest status of the epidemic using sources of UNAIDS, Geneva as well as those from India

2. The introduction/methods/results/discussions may be limited to the aspects which are most relevant to the study objectives.

3. The manuscript needs to detail the limitations as standard.

Reviewer #2: 0- The authors have presented findings of a very pertinent research, which could be a crucial strategy for preventing new HIV infections in Asia and Pacific, and of course India, where the study has been done.

1- However, a few points in the article needs to be looked into.

- About the study's objective, and research question: The authors put three statements

{

It aimed to understand the barriers and enablers in acceptance of PrEP as a method of HIV prevention among MSM and Transgender in India. This study also attempts to provide plausible solutions and measures that would be required before rolling out the PrEP programs in India, by understanding and tailoring the program to the needs and concerns of the TG and MSM. It attempts to capture the experience and needs for

different MSM typologies and TG in two culturally different contexts in India.

}

Each of these could be potential research questions, which would require appropriate study design to answer meaningfully. For example, strategies to best tailor for a future programmatic implementation in India, can not come from a qualitative study only, rather, would require a few demonstration projects to best answer that question.

The methods, and findings of the study are on the qualitative aspect of the possible larger work that was perhaps done. Hence, it is desirable that the authors may try to give a specific objective, and a specific research question for the work that follows, in line with methods and results.

2- Authors have integrated the findings of the focus group discussion, and individual interview data. While this is not totally unacceptable, but it is leaving avenues of confusion. FGDs are best to give a glimpse of the novelties that are arising, while IDIs can be best used to triangulate the findings of the FGDs. In some cases, repeat FGDs are done to further substantiate the findings and merging themes.

Hence, method section may have details on chronology of the FGDs and IDIs, presentation of the findings separately for FGD and IDIs, and then trying to see if they are indeed triangulating.

3- Sampling strategy: Authors have very superficially mentioned that 'a mix of convenience and purposive sampling' was used to select the participants for FGD. Qualitative study does not mean that sampling strategy is not useful, or can be noted in passing. Since, this study presents cumulative findings of Delhi and Bangalore, it becomes very confusing when one tries to imagine how the samples were approached, and what was the sampling frame.

Hence, the authors should a) clearly delineate what do they meant by 'mix of convenience and purposive'?

b) They should give a clear account of sampling strategy, by describing the sampling frame. c) They should give numbers of MSM sites and TG sites in Delhi and Bangalore respectively (if targeted intervention sites were approached), and then how did they proceed to drill down upto the level of individuals in each city, in each site. c) Also give information on whether they were enrolled by a NGO, and was their any differential enrolment between TG and MSM-> this will help readers to appreciate many differences between MS and TG with regards to PrEP.

4- Eligibility criteria: While authors have given detail account of MSM, TGnj and TGh definitions, they have not presented anywhere, how exactly did they identify them? a) This needs to be given. b) Also within a given study site (eg targeted intervention) how MSM, and TGs were defined, and approached needs to be given.

5- TGh and TGnh were not much described in results later on, so authors may see if this additional distinction is a must requirement in introduction section?

6. PLOS authors have the option to publish the peer review history of their article (what does this mean?). If published, this will include your full peer review and any attached files.

Reviewer #1: No

Reviewer #2: **Yes: **Dr. Partha Haldar, Centre for Community Medicine, All India Institute of Medical Sciences, New Delhi, India

---

## [Author Response · Author response to Decision Letter 0]

3 Mar 2022

Response to Reviewers

***

Table 2. Specific observations

Section Line number Page number (Manuscript) Comments Response to Reviwer Comments

Introduction 46 4 There are updates in HIV burden estimates in India. The author may like to review, update and reference appropriately.

 The suggested changes were incorporated and the citations are updated. 

 47 4 HIV prevalence is not an indicator to measure the impact of national/global AIDS response. The author might like to relate the impact with new infections, AIDS-related deaths, and survival and reference accordingly. 

 Agree, as suggested, the data on new infections and AIDS-related mortality were incorporated in the revised manuscript. 

 58 4 There are updates in HIV burden estimates in India. The author may like to review, update and reference appropriately.

 The most recent data on HIV prevalence among MSM and Transgender population were included and related (USAID and NACO) citations were updated. 

 60-62 4 The size of MSM population is different from the official estimates of Govt of India. The author may bring that perspective also to provide a more comprehensive picture to the reader. Agree. A statement indicating the estimation of MSM population in India is much lesser than the actual number and this is because of the challenges associated with disclosure of identity and the estimation challenges are presented in the revised manuscript. 

About the MSM and TG population in India 92-150 6-8 This whole section, while important in itself, appears redundant in the context of this manuscript. The Author may consider including the definition of MSM and TG used in this study in the methodology section and present this whole section about the MSM and TG in India as supplementary material. Agree. A definition of MSM and TG used in this study are presented in the methodology. The full section on “About the MSM and Transgender Community in India” is presented in the supplementary section. 

Methodology 170 9 The study employed convenience and purposive sampling methods. The author may like to describe the study settings in brief to provide a perspective to the readers. 

 We have made slight changes in the sampling methods and elaborated the study setting to provide a better perspective to the readers. 

 171-172 9 The Author may define the distinct types of MSM and TGs recruited in this study here.

 Yes, we have made the suggested changes and incorporated the relevant citations. 

 175 9 The manuscript not necessarily aims to present any findings by sub-typology of the MSM and TG. The author may consider focusing on the MSM and TG without any further reference to the specific sub-typologies. 

 Agree. The details of sub-typologies were removed and the findings now focus on MSM and TGs 

 177 10 May please put the interview guide in brief as a box either in the manuscript or put the detailed tool as supplementary material.

 The specific themes used in the interview guide were elaborated in this section. But adding a table would cut the flow of this section. However, If the journal permits, we could add the detailed tool as supplementary material. Thank you for this feedback. 

 187-188 10 Please mention ‘study objectives and methods’ as areas of the training. 

 Incorporated these changes in the revised manuscript. 

 191-198 10 This paragraph details the study settings. However, it would be good to summarize them here and provide the details in the supplementary material perhaps.

 We modified this paragraph and summarize the study setting in the revised manuscript. 

 222 12 The use of the word ‘paid’ may be reviewed as it breaks the flow of the sentence. 

 Sorry, this was a typo error, and the word ‘paid’ was removed in the revised manuscript. 

Results 230-595 12-30 1. The author may consider being a bit more judicious while presenting the quotes and thematic areas. Only the areas which are directly and significantly related to the manuscript objectives may be presented. This will help the manuscript to be more focused and reader-friendly. 

2. There is inconsistency in the presentation of the setting and speakers in the quotes. Sometimes, they have been presented by pseudonyms. Sometimes by the respondent ID. Sometimes, the locations have been mentioned. Sometimes, not. Consistency shall be ensured while 

the presentation of presenting the setting and speakers in the quotes

3. The results in line 298 appears to be stigmatising a sub-group as inference is being written without mentioning any quote directly referring to the same. At the same time, the findings here doesn't add any significant findings in the context of the manuscript objective. The author may consider removing this result. 

4. Consistency shall be established in results in different sections. In line 250, the results say both TG and MSMs had poor knowledge of PrEP. On the other hand, Table 2 specifically mentions as of TGs are well informed on PrEP. The author may like to review the results to ensure consistency in the presentation of results and concluding.

 Thank you for this valuable feedback. We revisited the results sections and modified the quotes wherever necessary. In some places, more than one quote was presented earlier and it has been slightly changed now. We tried to retain the most relevant quotes and also, some writeups in the results were also modified based on this feedback. 

Thank you for this important observation. As you suggested, we tried to present this more consistently. In the revised manuscript, we have included pseudonyms, participant sexual identity, age and geography. I think, this provides a better picture of the presented quotes and helps the readers to contextualize the informations. 

This paragraph has been removed in the revised manuscript. 

Your observations are correct. Line 250 is slightly modified now. In the revised manuscript, it says, “in both the sites, none of the MSM and TG people reported taking the PrEP ever”. After this sentence and also later in table 2, it was stated that “in both places, TGs displayed a better understanding of PrEP as compared to the MSM group”. This is the correct information. 

Discussion 631-633 32 1. The statement that respondents clamoured for PrEP is inconsistent with the study finding of low awareness about PrEP. The author may like to rework the sentences.

2. The manuscript does not present any limitations. It is important to mention the same to provide perspectives to the readers. The word ‘clamoured’ is replaced with the word ‘inclined’. Basically, it says that most participants are motivated towards PrEP considering the challenges they have been facing in terms of correct and consistent use of condoms as HIV prevention strategy. 

We presented the limitations in the revised manuscript. Thank you for this observation and feedback.

---

## [Decision Letter · Decision Letter 1]

7 Jun 2022

PONE-D-21-09058R1Acceptability of PrEP among MSM and Transgender Communities - qualitative findings from two metropolitan cities in IndiaPLOS ONE

Dear Dr. Ramanaik,

Thank you for submitting your manuscript to PLOS ONE. After careful consideration, we feel that it has merit but does not fully meet PLOS ONE’s publication criteria as it currently stands. Therefore, we invite you to submit a revised version of the manuscript that addresses the points raised during the review process. Unfortunately, one of the initial reviewers was not available to assess the revisions made, and thus a new reviewer was assigned. This reviewer has very helpfully offered line-by-line suggestions for revising the manuscript. In light of the overall evaluation that the manuscript required professional proofreading in order to be comprehensible to an international readership, following these detailed suggested will be extremely helpful in improving readability. The suggestion about the placement of the limitations, and adding the further limitation indicated, is also in keeping with academic standards. Please when you make these revisions, indicate the location iof each revision in the text; this will expedite the editorial review process.

We look forward to receiving your revised manuscript.

Kind regards,

Peter A. Newman, Ph.D

Academic Editor

PLOS ONE

Journal Requirements:

Reviewers' comments:

Reviewer's Responses to Questions

**Comments to the Author**

1. If the authors have adequately addressed your comments raised in a previous round of review and you feel that this manuscript is now acceptable for publication, you may indicate that here to bypass the “Comments to the Author” section, enter your conflict of interest statement in the “Confidential to Editor” section, and submit your "Accept" recommendation.

Reviewer #1: All comments have been addressed

Reviewer #3: (No Response)

2. Is the manuscript technically sound, and do the data support the conclusions?

Reviewer #1: Yes

Reviewer #3: Yes

3. Has the statistical analysis been performed appropriately and rigorously? 

Reviewer #1: N/A

Reviewer #3: N/A

4. Have the authors made all data underlying the findings in their manuscript fully available?

Reviewer #1: No

Reviewer #3: Yes

5. Is the manuscript presented in an intelligible fashion and written in standard English?

Reviewer #1: Yes

Reviewer #3: No

6. Review Comments to the Author

Reviewer #1: (No Response)

Reviewer #3: This paper provides qualitative data on the acceptability of PrEP in a diverse cohort of MSM (including those who have a westernised gay identity and those who are described by traditional Indian terms) and transgender women in two different settings in India. The participants in this study do not have access to PrEP, so they are responding to and intervention that has been described to them, rather than one they have personally used.

While I recognise that this paper has already undergone a round of review, I am recommending further revisions which I hope will add to the clarity of the paper and situate its findings more clearly in relation to other literature.

Please note that all the line numbers I reference are from the tracked changes version.

1. Firstly, I recommend professional copy editing for this manuscript, as there are numerous occasions where the wording choice is slightly jarring or the tense wrong. That said, I recognise that this is not always possible so I will endeavour to point out examples of wording choices and tenses.

2. I would strongly suggest that the umbrella term for the transgender respondents be changed to ‘transgender women’ (no capital ‘T’ needed for transgender). The use of ‘people’ initially made me think that both trans men and women – and possibly non-binary people – were included in the study, whereas it is all people with feminine identification (some who identify with the traditional term, hijra., and some who do not). If the authors feel it is inappropriate to use the term ‘women’ I bow to their superior understanding of the cultural context, but in my country when a trans person identifies as a woman it is considered respectful to call her a woman, regardless of what medical interventions she has or has not had.

3. Line 24, suggest ‘involving’ rather than ‘covering

4. Line 25 suggest ‘explored’ not ‘tired to explore

5. Line 32 – no semi colon after ‘acceptance’ – use comma

6. Line 39 – cut ‘makes a unique attempt to’ instead say ‘provides’

7. Line 42 – there;s a hanging ‘s’

8. Lines 67-70 ‘lesser’ should be less, and this line needs a citation

9. Line 78 – replace ‘sexual orientation’ with ‘gender identity

10. Line 172 – cut ‘was’

11. Line 175 ‘participants’’ – needs apostrophe

12. Lines 188-90 I suggest removing the phrase ‘typically have more effeminate mannerisms’ altogether, but if the author strongly disagree, then change the word ‘effeminate’ – which is pejorative – to ‘feminine’

13. Line 191. As I have recommended above, this paper will work better cross culturally if the term ‘transgenderer women’ is used throughout. At line 191, where there is a definition given, this could include ‘transgender women were reported male at birth but self-identify as women or feminine’

14. Line 203 – I suggest cutting the line that begins ‘A participatory approach’ – as this is usually understood to mean an approach where there is a stronger partnership between the researcher community and the researchers, from the inception of the project to the dissemination of results. What is described here is a considered approach that aims to ‘give back’ to research participants, which is laudable, but it is different to a participatory approach.

15. Line 225 no hyphen between two and days

16. Line 226 ‘included’ not ‘includes’

17. Line 265 – pseudonyms, not name

18. Line 226 can delete the line that begins ‘this additional consideration’ – extraneous, this is clear from the preceding line.

19. Line 302 – replace ‘comprehend’ with ‘misunderstood’

20. Line 303 – ‘the first introduction to THE CONCEPT OF PrEP’ – add the capitatlised words

21. Line 311 - After providing a briefing on PrEP (add italicised words)

22. Line 312 – delete ‘have’ , making it ‘the participants also’

23. Line 315 – replace ‘communities’ with ‘respondents’ or ‘participants’

24. Line 317 – ‘find’ should be ‘found’

25. Line 326 – The participants express a view that PrEP might reduce the need for HIV testing – was this misapprehension addressed with participants in the information provided, given that frequent testing remains important?

26. Line 327 – avoid the use of ‘shared’ use ‘reported’ or ‘suggested’

27. Line 341- text marked deleted – in this section a range of participants talk about sex where the lack of protection is forced by insertive partners, which reveal the profound power disparity between feminine or receptive sex partners and insertive partners. It seems important to me to keep some of this in the paper. I recommend cutting the section on recommendation regarding the general community, (484onwards) but keeping the section that relates to experiences of violence.

28. Line 455 – Don’t use ‘deviation’ – say ‘diversity’

29. Line 475 – ‘reported’ not ‘shared’

30. Line 478 – change ‘deteriorate’ to ‘exacerbate’

31. Line 487 change ‘are not’ to ‘is not’

32. Line 537 – it is not necessary to start taking PrEP 6 days before risk exposure (See data on ‘on demand’ PrEP by Molina) – this should be acknowledged in the text somewhere

33. Line 593 – ‘social pariah’s is a loaded term. How about ‘people of lower social status’ instead?

34. Line 660 – change ‘categorise’ to ‘distinguish’

35. Line 695 – rather than ;’nclined towards’ – ‘advocated strongly for’ PrEP

36. Line 697 – apostrophe in participant’s should be participants’

37. Line 713 ‘their traveling’ should be ‘the mobile’

38. Line 716-17 – this line takes the Australian study out of context (I am an author of it). The study was in the context of COVID-related changes and not broadly applicable to this context, excepting that is some instances people prefer to take PrEP medication irregularly, particularly if sex is irregular.

39. The paragraph beginning at line &31 should also address the misconception from the results that PrEP could reduce the need for HIV testing.

40. Line 745 – replace ‘This’ with ‘Discriminatory attitudes displayed by healthcare providers’

41. Line 755 ‘raised’ rather than ‘brought forth’

42. Line 805 – move the limitations to the end of the discussion, and commence with, “this study has some limitations’. After the line, ‘None of the participants have ever used PrEP’ add – ‘and they were dependent upon explanations from the research team to understand how it worked’

7. PLOS authors have the option to publish the peer review history of their article (what does this mean?). If published, this will include your full peer review and any attached files.

Reviewer #1: **Yes: **Pradeep Kumar

Reviewer #3: **Yes: **Bridget Haire

---

## [Author Response · Author response to Decision Letter 1]

2 Aug 2022

Response to editor comments:

PONE-D-21-09058R1

Acceptability of PrEP among MSM and Transgender Communities - qualitative findings from two metropolitan cities in India

PLOS ONE

Dear Dr. Ramanaik,

Thank you for submitting your manuscript to PLOS ONE. After careful consideration, we feel that it has merit but does not fully meet PLOS ONE’s publication criteria as it currently stands. Therefore, we invite you to submit a revised version of the manuscript that addresses the points raised during the review process.

Unfortunately, one of the initial reviewers was not available to assess the revisions made, and thus a new reviewer was assigned. This reviewer has very helpfully offered line-by-line suggestions for revising the manuscript. In light of the overall evaluation that the manuscript required professional proofreading in order to be comprehensible to an international readership, following these detailed suggested will be extremely helpful in improving readability. The suggestion about the placement of the limitations, and adding the further limitation indicated, is also in keeping with academic standards. Please when you make these revisions, indicate the location of each revision in the text; this will expedite the editorial review process.

- We are grateful to the editorial team and the reviewers for their constructive feedback. This has greatly helped us to improve the quality of this paper. We have addressed each of the suggestions made by Reviewer 3 have also completed a professional proofreading in alignment with international readership. The limitation paragraph has been moved next to discussion section and all other suggested changes have been incorporated. All changes are made in track-change mode to expedite the review process. 

- We have uploaded the above mentioned items as attachments along with the revised manuscript. 

We look forward to receiving your revised manuscript.

Kind regards,

Peter A. Newman, Ph.D

Academic Editor

PLOS ONE

Journal Requirements:

Response to reviewers’ comments:

Reviewer's Responses to Questions

Comments to the Author

1. If the authors have adequately addressed your comments raised in a previous round of review and you feel that this manuscript is now acceptable for publication, you may indicate that here to bypass the “Comments to the Author” section, enter your conflict of interest statement in the “Confidential to Editor” section, and submit your "Accept" recommendation.

Reviewer #1: All comments have been addressed

Reviewer #3: (No Response)

2. Is the manuscript technically sound, and do the data support the conclusions?

Reviewer #1: Yes

Reviewer #3: Yes

3. Has the statistical analysis been performed appropriately and rigorously? 

Reviewer #1: N/A

Reviewer #3: N/A

4. Have the authors made all data underlying the findings in their manuscript fully available?

Reviewer #1: No

Reviewer #3: Yes

5. Is the manuscript presented in an intelligible fashion and written in standard English?

Reviewer #1: Yes

Reviewer #3: No

6. Review Comments to the Author

Reviewer #1: (No Response)

- Thank you for your valuable feedback and the time spent to review this paper

Reviewer #3: This paper provides qualitative data on the acceptability of PrEP in a diverse cohort of MSM (including those who have a westernised gay identity and those who are described by traditional Indian terms) and transgender women in two different settings in India. The participants in this study do not have access to PrEP, so they are responding to and intervention that has been described to them, rather than one they have personally used.

While I recognise that this paper has already undergone a round of review, I am recommending further revisions which I hope will add to the clarity of the paper and situate its findings more clearly in relation to other literature.

- Thanks very much for your valuable feedback and the comments. We have tried our best to respond to all your comments. 

Please note that all the line numbers I reference are from the tracked changes version.

1. Firstly, I recommend professional copy editing for this manuscript, as there are numerous occasions where the wording choice is slightly jarring or the tense wrong. That said, I recognise that this is not always possible so I will endeavour to point out examples of wording choices and tenses.

- Your feedback on the language is very helpful. We have addressed your feedback on the language and have also completed a professional proof-reading for language proficiency. 

2. I would strongly suggest that the umbrella term for the transgender respondents be changed to ‘transgender women’ (no capital ‘T’ needed for transgender). The use of ‘people’ initially made me think that both trans men and women – and possibly non-binary people – were included in the study, whereas it is all people with feminine identification (some who identify with the traditional term, hijra., and some who do not). If the authors feel it is inappropriate to use the term ‘women’ I bow to their superior understanding of the cultural context, but in my country when a trans person identifies as a woman it is considered respectful to call her a woman, regardless of what medical interventions she has or has not had.

- We have used the terminology ‘transgender community’ or ‘transgender communities’ or transgender individuals which is the currently accepted terminology at an international level. We think this is more appropriate and acceptable in the local context. We would prefer to use this terminology instead of the term specifically referring to the community as ‘women’ or ‘men’. 

3. Line 24, suggest ‘involving’ rather than ‘covering

- Agree. We have incorporated this change. 

4. Line 25 suggest ‘explored’ not ‘tired to explore

- Agree. We have incorporated this change 

5. Line 32 – no semi colon after ‘acceptance’ – use comma

- Agree. We have incorporated this change 

6. Line 39 – cut ‘makes a unique attempt to’ instead say ‘provides’

- Agree. We have incorporated this change 

7. Line 42 – there; s a hanging ‘s’

- Agree. We have incorporated this change

8. Lines 67-70 ‘lesser’ should be less, and this line needs a citation

- Agree. We have incorporated this change 

9. Line 78 – replace ‘sexual orientation’ with ‘gender identity

- Agree. We have incorporated this change 

10. Line 172 – cut ‘was’

- Agree. We have incorporated this change 

11. Line 175 ‘participants’’ – needs apostrophe

- Agree. We have incorporated this change 

12. Lines 188-90 I suggest removing the phrase ‘typically have more effeminate mannerisms’ altogether, but if the author strongly disagree, then change the word ‘effeminate’ – which is pejorative – to ‘feminine’

- Agree. We have incorporated this change 

13. Line 191. As I have recommended above, this paper will work better cross culturally if the term ‘transgender women’ is used throughout. At line 191, where there is a definition given, this could include ‘transgender women were reported male at birth but self-identify as women or feminine’

- Agree. We have used transgender community. 

14. Line 203 – I suggest cutting the line that begins ‘A participatory approach’ – as this is usually understood to mean an approach where there is a stronger partnership between the researcher community and the researchers, from the inception of the project to the dissemination of results. What is described here is a considered approach that aims to ‘give back’ to research participants, which is laudable, but it is different to a participatory approach.

- Agree. We have incorporated this change 

15. Line 225 no hyphen between two and days

- Agree. We have incorporated this change 

16. Line 226 ‘included’ not ‘includes’

- Agree. We have incorporated this change 

17. Line 265 – pseudonyms, not name

- Agree. We have incorporated this change 

18. Line 266 can delete the line that begins ‘this additional consideration’ – extraneous, this is clear from the preceding line.

- Agree. We have incorporated this change

19. Line 302 – replace ‘comprehend’ with ‘misunderstood’

- Agree. We have incorporated this change 

20. Line 303 – ‘the first introduction to THE CONCEPT OF PrEP’ – add the capitalised words

- Agree. We have incorporated this change

21. Line 311 - After providing a briefing on PrEP (add italicised words)

- Agree. We have incorporated this change

22. Line 312 – delete ‘have’, making it ‘the participants also’

- Agree. We have incorporated this change 

23. Line 315 – replace ‘communities’ with ‘respondents’ or ‘participants’

- Agree. We have incorporated this change

24. Line 317 – ‘find’ should be ‘found’

- Agree. We have incorporated this change 

25. Line 326 – The participants express a view that PrEP might reduce the need for HIV testing – was this misapprehension addressed with participants in the information provided, given that frequent testing remains important?

- Yes. After every session, the misapprehensions and concerns that were raised by the participants were addressed and explained. One of the research team members was a medical doctor. He clarified questions related to HIV testing and other biomedical concerns. 

26. Line 327 – avoid the use of ‘shared’ use ‘reported’ or ‘suggested’

- Agree. We have incorporated this change

27. Line 341- text marked deleted – in this section a range of participants talk about sex where the lack of protection is forced by insertive partners, which reveal the profound power disparity between feminine or receptive sex partners and insertive partners. It seems important to me to keep some of this in the paper. I recommend cutting the section on recommendation regarding the general community, (384onwards) but keeping the section that relates to experiences of violence.

- We agree with your suggestion and have retained the section suggested by you. 

- The community’s recommendations on providing PrEP to other high-risk communities is relevant in the context of reducing HIV transmission. We consider that this is an important piece of information for policy makers to consider. We have therefore retained this recommendation as well. 

28. Line 455 – Don’t use ‘deviation’ – say ‘diversity’

- Agree. We have incorporated this change 

29. Line 475 – ‘reported’ not ‘shared’

- Agree. We have incorporated this change

30. Line 478 – change ‘deteriorate’ to ‘exacerbate’

- Agree. We have incorporated this change 

31. Line 487 change ‘are not’ to ‘is not’

- Agree. We have incorporated this change 

32. Line 537 – it is not necessary to start taking PrEP 6 days before risk exposure (See data on ‘on demand’ PrEP by Molina) – this should be acknowledged in the text somewhere

- Thank you for sharing this new information. We have used the standard information that was available at the time data was collected. 

33. Line 593 – ‘social pariah’s is a loaded term. How about ‘people of lower social status’ instead?

- Agree. We have incorporated this change 

34. Line 660 – change ‘categorise’ to ‘distinguish’

- Agree. We have incorporated this change 

35. Line 695 – rather than ;’inclined towards’ – ‘advocated strongly for’ PrEP

- Agree. We have incorporated this change 

36. Line 697 – apostrophe in participant’s should be participants’

- Agree. We have incorporated this change 

37. Line 713 ‘their traveling’ should be ‘the mobile’

- Agree. We have incorporated this change 

38. Line 716-17 – this line takes the Australian study out of context (I am an author of it). The study was in the context of COVID-related changes and not broadly applicable to this context, excepting that is some instances people prefer to take PrEP medication irregularly, particularly if sex is irregular.

- Agree. Thank you for this clarification. 

39. The paragraph beginning at line &731 should also address the misconception from the results that PrEP could reduce the need for HIV testing.

- Agree. We have incorporated this change 

40. Line 745 – replace ‘This’ with ‘Discriminatory attitudes displayed by healthcare providers’

- Agree. We have incorporated this change 

41. Line 755 ‘raised’ rather than ‘brought forth’

- Agree. We have incorporated this change 

42. Line 805 – move the limitations to the end of the discussion, and commence with, “this study has some limitations’. After the line, ‘None of the participants have ever used PrEP’ add – ‘and they were dependent upon explanations from the research team to understand how it worked’

- Agree. We have incorporated this change 

7. PLOS authors have the option to publish the peer review history of their article (what does this mean?). If published, this will include your full peer review and any attached files.

Do you want your identity to be public for this peer review? For information about this choice, including consent withdrawal, please see our Privacy Policy.

Reviewer #1: Yes: Pradeep Kumar

Reviewer #3: Yes: Bridget Haire

---

## [Editor Report · Decision Letter 2]

17 Aug 2022

PONE-D-21-09058R2Acceptability of PrEP among MSM and Transgender Communities - qualitative findings from two metropolitan cities in IndiaPLOS ONE

Dear Dr. Ramanaik,

Thank you for submitting your manuscript to PLOS ONE. After careful consideration, we feel that it has merit but does not fully meet PLOS ONE’s publication criteria as it currently stands. Therefore, we invite you to submit a revised version of the manuscript that addresses the points raised during the review process.

The authors have largely responded to the reviewer comments; however, in so doing they have introduced new errors into the manuscript that prevent it from being accepted in its current form.

1) The revisions are largely responsive to the reviewers' comments. However, this added statement in the Discussion section is simply not true; plus the citations the authors use in fact do the opposite of what they say. This is more problematic since this overstatement and sweeping characterization of many studies about trans people in India is used to claim the present study as unique. In fact, the authors need to cite these studies when they present their own results, and indicate various factors that have in fact already been identified in the published literature about PrEP use among MSM and among transgender people—not mischaracterize existing studies to claim the uniqueness of their own. 

Strangely, these 2 studies (13, 14) are cited as conflating MSM and trans people; but to the contrary, they demonstrate separate inquiries about PrEP in each population:

“In recent years, there has been an increase in the number of studies on that broach the subject of PrEP acceptability among MSM in India (13, 14). These studies do not distinguish transgender communities from the MSM population.”

13. Chakrapani V, Newman PA, Shunmugam M, Mengle S, Varghese J, Nelson R, et al. Acceptability of HIV Pre-Exposure Prophylaxis (PrEP) and Implementation Challenges Among Men Who Have Sex with Men in India: A Qualitative Investigation. AIDS Patient Care STDS. 2015;29(10):569-77.

14. Chakrapani V, Shunmugam M, Rawat S, Baruah D, Nelson R, Newman PA. Acceptability of HIV Pre-Exposure Prophylaxis Among Transgender Women in India: A Qualitative Investigation. AIDS Patient Care STDS. 2020;34(2):92-8.

2) The following statement in the abstract is erroneous. The authors claim to address these two populations as distinct, but then report as if HIV prevalence is the same; however, as the authors likely know, this is untrue. The sentence either needs to be re-worded or the different prevalence estimates among these populations indicated.

“Men who have sex with men (MSM) and transgender communities in India have 24-times higher HIV prevalence than the general population.”

Also, this information in the abstract conflicts with what is written in the Introduction:

“HIV prevalence among MSM (2.7%) and transgender community people (3.1%) is about 12 to 14 fold higher than the general population (0.22%) (4)

Both of these statements cannot be correct. 24 times higher? Or 12-14 fold higher?

3) I accept the authors’ response to Reviewer #3, and their use of the term transgender individual is fine, as is transgender community when not referring to an individual. However, the authors still move back and forth between capitalizing “Transgender” and using lower case “transgender”. Note that MSM is capitalized because it is an acronym; the word transgender should not be capitalized, as correctly indicated by the reviewer. It is now presented randomly in both formats, which is incorrect.

The problem with on and off capitalization of transgender starts in the abstract, but is repeated throughout the manuscript:

For ex.: Line 33: “However, when informed about PrEP, both MSM and Transgender communities expressed their willingness to use PrEP...."

But in the previous sentence, it is correctly written as transgender: “This study found that awareness and use of PrEP were minimal among the MSM and transgender communities.”

Then on p. 6: “There are very few studies on the acceptability of PrEP among MSM and transgender in India”

This is written without “individuals” or “communities”, just among “transgender”, which is incorrect.

4) DISCUSSION: This statement in the first sentence below does not logically correspond to the two following sentences. How can respondents “strongly advocate for PrEP” when “awareness was slight” and “knowledge of PrEP received mixed responses”? At least the first sentence needs to be re-written if you mean that AFTER they learned about PrEP, they were strongly supportive….

Also, there should be no apostrophe after participants in the 3rd sentence.

“Due to persistent risks associated with inconsistent use of condoms, most respondents have advocated strongly for PrEP. Very few among the study population had awareness of PrEP and its uses. Awareness was slightly higher more among the transgender communities. For many of the participants’ the first introduction to PrEP was through this study and knowledge of PrEP received mixed responses."

Additionally, it is important that the authors change this statement to indicate that it is not only due to “inconsistent condom use” that PrEP can be helpful for these communities. This statement implies that all sexual encounters are consensual and all HIV risk is an individual choice, as if to lay blame on individuals. In fact, several studies indicate high rates of sexual violence against both MSM and transgender people in India, and how that impacts on condom use--in addition to depression, alcohol use, etc. It is not merely choosing or not to use a condom. The following article shows that violence victimization, for one, is associated with lower rates of condom use. It also shows separate analyses among MSM and among transgender people, which provides yet another example that does not support the authors' claim that all previous HIV prevention research with transgender people in India fails to distinguish them from MSM. While their claim is true to an extent, they need to tone it down AND indicate examples of work (including the citation below, and references 13 and 14 above) that in fact has differentiated clearly between these two populations:

Venkatesan Chakrapani, Peter A. Newman, Murali Shunmugam, Carmen H. Logie & Miriam Samuel (2017) Syndemics of depression, alcohol use, and victimisation, and their association with HIV-related sexual risk among men who have sex with men and transgender women in India, Global Public Health, 12:2, 250-265, DOI: 10.1080/17441692.2015.1091024

5) The authors are unclear in their response about having the manuscript professionally proofread by someone fluent in English. While improved, it includes many errors in grammar and style, including formatting of citations. Since this journal does not have a proofreading stage, the authors must take this step themselves. For ex., in the first paragraph of the Introduction:

This sentence is incorrect. The way this is cited is wrong: " As per the recently released (2019) HIV estimation report 2019,..."

This sentence is missing a close parenthesis after (FSW … : Key populations including female sex workers (FSW, men who have sex with men (MSM), transgender community and persons who inject drugs (PWID)…

These comments present only a few of the instances showing the need for proofreading and corrections. The entire manuscript must be professionally proofread and corrected.

We look forward to receiving your revised manuscript.

Kind regards,

Peter A. Newman, Ph.D

Academic Editor

PLOS ONE
---

## [Author Response · Author response to Decision Letter 2]

24 Jan 2023

ONE-D-21-09058R2

Acceptability of PrEP among MSM and Transgender Communities - qualitative findings from two metropolitan cities in India

PLOS ONE

Dear Dr. Ramanaik,

Thank you for submitting your manuscript to PLOS ONE. After careful consideration, we feel that it has merit but does not fully meet PLOS ONE’s publication criteria as it currently stands. Therefore, we invite you to submit a revised version of the manuscript that addresses the points raised during the review process.

The authors have largely responded to the reviewer comments; however, in so doing they have introduced new errors into the manuscript that prevent it from being accepted in its current form.

1) The revisions are largely responsive to the reviewers' comments. However, this added statement in the Discussion section is simply not true; plus the citations the authors use in fact do the opposite of what they say. This is more problematic since this overstatement and sweeping characterization of many studies about trans people in India is used to claim the present study as unique. In fact, the authors need to cite these studies when they present their own results, and indicate various factors that have in fact already been identified in the published literature about PrEP use among MSM and among transgender people—not mischaracterize existing studies to claim the uniqueness of their own. 

Strangely, these 2 studies (13, 14) are cited as conflating MSM and trans people; but to the contrary, they demonstrate separate inquiries about PrEP in each population:

“In recent years, there has been an increase in the number of studies on that broach the subject of PrEP acceptability among MSM in India (13, 14). These studies do not distinguish transgender communities from the MSM population.”

13. Chakrapani V, Newman PA, Shunmugam M, Mengle S, Varghese J, Nelson R, et al. Acceptability of HIV Pre-Exposure Prophylaxis (PrEP) and Implementation Challenges Among Men Who Have Sex with Men in India: A Qualitative Investigation. AIDS Patient Care STDS. 2015;29(10):569-77.

14. Chakrapani V, Shunmugam M, Rawat S, Baruah D, Nelson R, Newman PA. Acceptability of HIV Pre-Exposure Prophylaxis Among Transgender Women in India: A Qualitative Investigation. AIDS Patient Care STDS. 2020;34(2):92-8.

Agree. We have incorporated the changes accordingly. These studies cover two different groups and are pioneer studies in investigating about PrEP acceptability in transgender women and MSMs separately in India. 

Earlier the statement tried to highlight that often there is ambiguity and over representation of transgender by including MSM, transgender men, transgender women and female sex workers under one umbrella. However, particularly these two studies clearly inquire about two different groups. Changes have been accordingly.

2) The following statement in the abstract is erroneous. The authors claim to address these two populations as distinct, but then report as if HIV prevalence is the same; however, as the authors likely know, this is untrue. The sentence either needs to be re-worded or the different prevalence estimates among these populations indicated.

“Men who have sex with men (MSM) and transgender communities in India have 24-times higher HIV prevalence than the general population.”

Also, this information in the abstract conflicts with what is written in the Introduction:

“HIV prevalence among MSM (2.7%) and transgender community people (3.1%) is about 12 to 14 fold higher than the general population (0.22%) (4)

Both of these statements cannot be correct. 24 times higher? Or 12-14 fold higher?

Agree, have incorporated the correct estimation. It is around 15-17 fold higher. Also, the figures for MSM have been updated in the introduction, referring to the latest data from HIV surveillance report 2021.

https://www.unaids.org/en/regionscountries/countries/india (same reference) only numbers updated

3) I accept the authors’ response to Reviewer #3, and their use of the term transgender individual is fine, as is transgender community when not referring to an individual. However, the authors still move back and forth between capitalizing “Transgender” and using lower case “transgender”. Note that MSM is capitalized because it is an acronym; the word transgender should not be capitalized, as correctly indicated by the reviewer. It is now presented randomly in both formats, which is incorrect.

The problem with on and off capitalization of transgender starts in the abstract, but is repeated throughout the manuscript:

For ex.: Line 33: “However, when informed about PrEP, both MSM and Transgender communities expressed their willingness to use PrEP...."

But in the previous sentence, it is correctly written as transgender: “This study found that awareness and use of PrEP were minimal among the MSM and transgender communities.”

Agree, incorporated the changes, made it uniform as ‘transgender communities’ or ‘transgender individual’ as per the context

Then on p. 6: “There are very few studies on the acceptability of PrEP among MSM and transgender in India”

This is written without “individuals” or “communities”, just among “transgender”, which is incorrect.

Agree. We have incorporated the changes, added ‘communities’ while referring to transgender community

4) DISCUSSION: This statement in the first sentence below does not logically correspond to the two following sentences. How can respondents “strongly advocate for PrEP” when “awareness was slight” and “knowledge of PrEP received mixed responses”? At least the first sentence needs to be re-written if you mean that AFTER they learned about PrEP, they were strongly supportive….

Also, there should be no apostrophe after participants in the 3rd sentence.

Agree, we stand corrected and have made the changes.

“Due to persistent risks associated with inconsistent use of condoms, most respondents have advocated strongly for PrEP. Very few among the study population had awareness of PrEP and its uses. Awareness was slightly higher more among the transgender communities. For many of the participants’ the first introduction to PrEP was through this study and knowledge of PrEP received mixed responses."

Additionally, it is important that the authors change this statement to indicate that it is not only due to “inconsistent condom use” that PrEP can be helpful for these communities. This statement implies that all sexual encounters are consensual and all HIV risk is an individual choice, as if to lay blame on individuals. In fact, several studies indicate high rates of sexual violence against both MSM and transgender people in India, and how that impacts on condom use--in addition to depression, alcohol use, etc. It is not merely choosing or not to use a condom. The following article shows that violence victimization, for one, is associated with lower rates of condom use. It also shows separate analyses among MSM and among transgender people, which provides yet another example that does not support the authors' claim that all previous HIV prevention research with transgender people in India fails to distinguish them from MSM. While their claim is true to an extent, they need to tone it down AND indicate examples of work (including the citation below, and references 13 and 14 above) that in fact has differentiated clearly between these two populations:

Venkatesan Chakrapani, Peter A. Newman, Murali Shunmugam, Carmen H. Logie & Miriam Samuel (2017) Syndemics of depression, alcohol use, and victimisation, and their association with HIV-related sexual risk among men who have sex with men and transgender women in India, Global Public Health, 12:2, 250-265, DOI: 10.1080/17441692.2015.1091024

Agree. As correctly pointed out ‘inconsistent condom use’ is only one of the factors but more importantly experiences of violence put the MSM and TG communities at a greater risk of HIV infection. This is added in the text and another research article (Shaw et. Al 2012) has been referred to inform the same, in addition to Chakrapani’s paper (2017).

5) The authors are unclear in their response about having the manuscript professionally proofread by someone fluent in English. While improved, it includes many errors in grammar and style, including formatting of citations. Since this journal does not have a proofreading stage, the authors must take this step themselves. For ex., in the first paragraph of the Introduction:

This sentence is incorrect. The way this is cited is wrong: " As per the recently released (2019) HIV estimation report 2019,..."

Agree. We have corrected the sentence

This sentence is missing a close parenthesis after (FSW … : Key populations including female sex workers (FSW, men who have sex with men (MSM), transgender community and persons who inject drugs (PWID)…

Agree. We have corrected the sentence

These comments present only a few of the instances showing the need for proofreading and corrections. The entire manuscript must be professionally proofread and corrected.

We look forward to receiving your revised manuscript.

Kind regards,

Peter A. Newman, Ph.D

Academic Editor

PLOS ONE
---

## [Editor Report · Decision Letter 3]

7 Feb 2023

Acceptability of PrEP among MSM and transgender communities - qualitative findings from two metropolitan cities in India

PONE-D-21-09058R3

Dear Dr. Ramanaik,

We’re pleased to inform you that your manuscript has been judged scientifically suitable for publication and will be formally accepted for publication once it meets all outstanding technical requirements.

Kind regards,

Peter A. Newman, Ph.D

Academic Editor

PLOS ONE
---

## [Editor Report · Acceptance letter]

10 Feb 2023

PONE-D-21-09058R3 

Acceptability of PrEP among MSM and transgender communities - qualitative findings from two metropolitan cities in India 

Dear Dr. Ramanaik:

I'm pleased to inform you that your manuscript has been deemed suitable for publication in PLOS ONE. Congratulations! Your manuscript is now with our production department. 

Kind regards, 

on behalf of

Dr. Peter A. Newman 

Academic Editor

PLOS ONE